# Fast and Generalized DeepFake Detector Through Feature Space Transformation

## Abstract

The current state-of-the-art DeepFake or manipulated image detection algorithms are not generalized against an unseen database, manipulation types, and image degradation due to compression. Existing literature shows different input transformations can boost the detection performance of deepfake detection algorithms. However, these algorithms only transform the spatial pixel values with the hope that the transformation will help in learning a linearly separable decision boundary. The transformation of a 2D volume containing millions of pixel values is computationally complex and on top of that, the amalgamation with the original image further increases the computational complexity. The proposed algorithm utilizes the concept of transformation; however, the transformation of a feature space that is 1-D and compact representation of an image. The transformed representation is then used to calculate the discriminative feature maps used for the binary classification as real or altered images. Extensive experimentation on multiple databases under several unconstrained settings establishes the effectiveness of the proposed algorithm and its desirability in the current era. Under each set, the proposed algorithm achieves state-of-the-art detection performance on Face Forensics++ and Celeb-DF databases. The proposed algorithm is almost 'parameter-free' and achieves its two-fold aim of giving a robust detection algorithm and an energy-saving medium.

## 1 Introduction

The presence of fake media on 'any' social media platforms has created havoc and made it hard to establish the authenticity of digital multimedia content. Due to this, it is now extremely dangerous to blindly trust the content on social media platforms and react to that. One such reason for such mistrust of social media content is the wide availability of DeepFake videos (Li et al., 2023b; Wang et al., 2023; Liang et al., 2022). These deepfake videos can be used for many purposes ranging from harassment to personal gain whether including monitory or achieving any post (WION, 2022). At the beginning of DeepFake, the face of celebrities was replaced with the face of pornstars in a video (Harwell, 2018). However, several recent incidents claim the importance of identifying which digital data is real and which is fake because it can severely impact any individual. A few such examples recently came into the picture are (i) the deepfake video of Ukraine's president telling his army to surrender (Telegraph, 2022), (ii) a candidate in the USA 2020 election claimed to be diagnosed with severe cancer and request the public to not vote for him as he is unfit for the post because of health issues (Cook, 2020), and (iii) deepfake audio is used to rob millions of dollars (Good, 2021). The problem became even more complicated because not only the sophisticated machine learning networks publicly available but simple mobile applications are now available which can easily be used for such malicious manipulations by a novice user (Agarwal et al., 2019a; Agarwal et al., 2017; Agarwal et al., 2021b). Therefore, the correct identification of these manipulated videos is not only important to build trust in society but also to make positive progress toward the advancement of artificial intelligence.

By looking at the severity of the problem, several research efforts have started to detect deepfake videos. The detection algorithms can be broadly divided into handcrafted plus machine learning classifiers and deep neural network data-driven algorithms. In one of the early works, (Agarwal et al., 2017a) proposed a novel feature engineering algorithm to highlight the subtle moiré patterns in the face swap videos. Other image

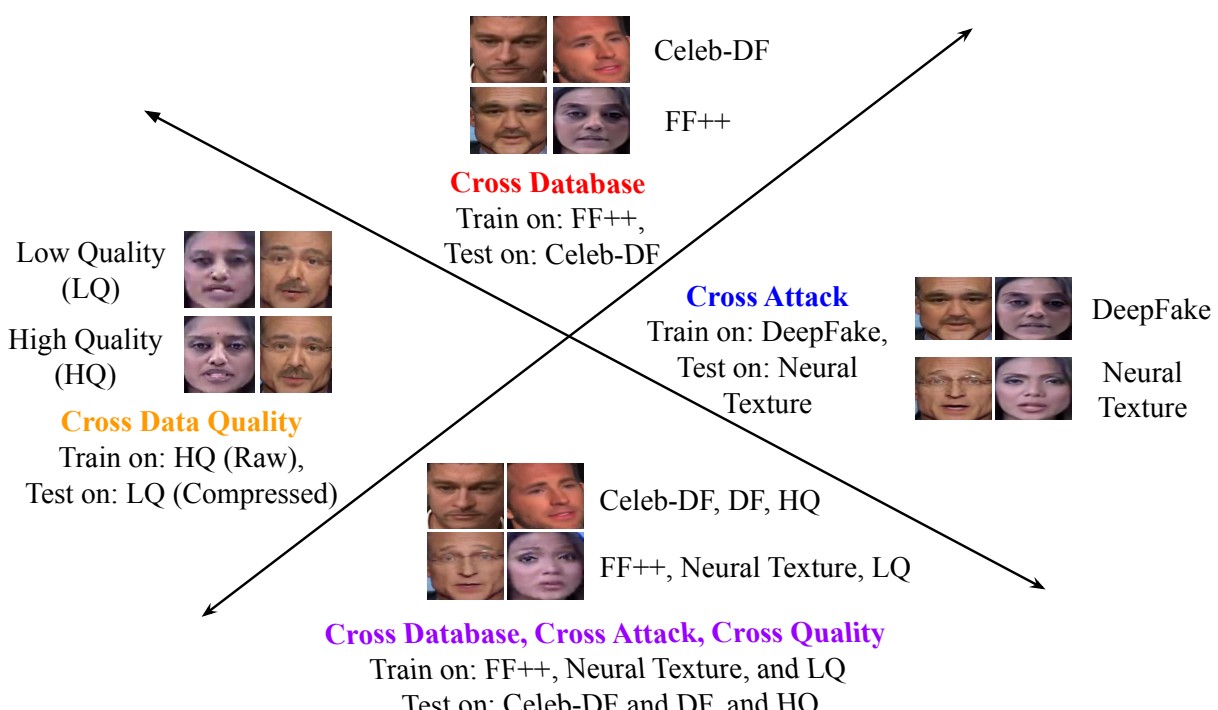

Figure 1: Missing challenges in existing image manipulation detection which we are trying to solve for a practical and robust defense against the serious real-world threat of the time. (i) cross-database, (ii) cross-attack, (iii) cross-data quality, and (iv) cross each possible variation (i.e., database, attack, and data quality).

feature features algorithm used for deepfake detection are: eye color and missing reflections (Matern et al., 2019), 3D head pose (Yang et al., 2019), facial movements (Baltrusaitis et al., 2018), and image artifacts (Raghavendra et al., 2017; Zhang et al., 2018). (Zhao et al., 2021c) and (Nirkin et al., 2021) have proposed source image features and face contextual information extraction networks for deepfake detection. The deepfake detection network proposed by (Zhao et al., 2021a) uses the multi-attention convolution network consisting of spatial attention heads and textural feature enhancement block. (Agarwal et al., 2021a) have proposed the generalized convolutional network architecture utilizing two branches consisting of raw images and transformed images and introduced the cross-stitch connections to transfer knowledge among layers of two branches (Agarwal et al., 2021a). (Zhou & Lim, 2021) have utilized both audio and video discrepancies for the detection of deepfake videos. DSP-FWA (Li et al., 2020c), Face X-ray (Li et al., 2020a), and PCL + I2G (Zhao et al., 2021c) proposed the boundaries in the facial regions which possibly exist due to the swapping of two faces. The above category of research algorithms possess two important characteristics: (i) either they are computationally efficient or (ii) yield state-of-the-art detection performance on multiple datasets. However, there is one catch here, the SOTA performance reported is usually obtained in a scenario where the training and testing images/videos belong to the same database and/or same attack type.

Henceforth, Figure 1 shows the motivation and desirability of the proposed research in the current era of fake examples surfacing over every possible corner of life. The existing defense algorithms are lacking these challenging scenarios while developing robust defense which makes them still not ready for practical deployment. The proposed research tackles all these challenges to build a secure and deployable detection algorithm. To summarize:

- A novel DeepFake detection algorithm is presented by utilizing an amalgamation of a convolution neural network, feature transformation network, and a traditional machine learning classifier;

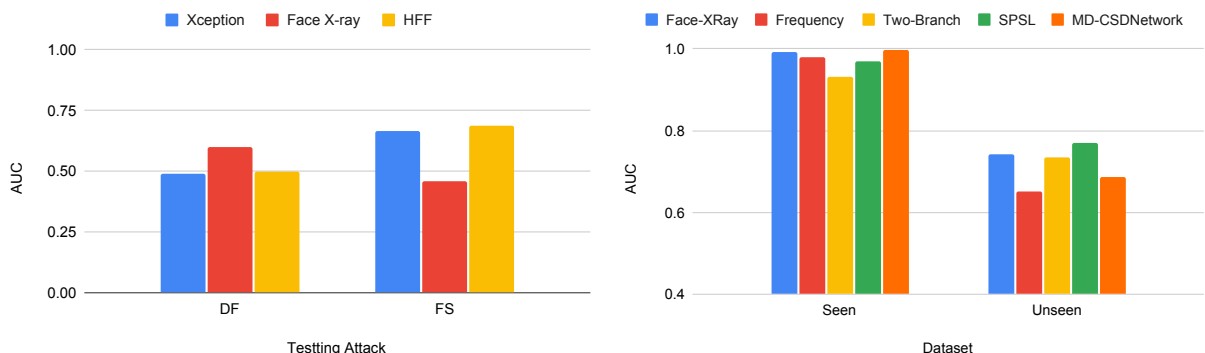

Figure 2: Critical areas where significant attention is needed in the deepfake detection research. The left chart shows the limitations of existing research published in top-tier venues under the cross and in the same attack setting. The AUC of three recent algorithms namely Face X-ray (Li et al., 2020a), XceptionNet, and High-Frequency (Luo et al., 2021) on seen attacks (Deepfake and face swap) is at least 0.981 which is at max 0.685 in unseen attack setting. Similarly, the right pie chart shows the bottleneck of the existing algorithms in terms of seen and unseen dataset training-testing. The performance of Face X-ray (Li et al., 2020a), MD-CSDNet (Agarwal et al., 2021a), SPSL (Liu et al., 2021a), Two branches (Masi et al., 2020), and Frequency (Qian et al., 2020) in the seen setting is at least 0.9318 whereas, in the unseen dataset, it shows the true generalization weakness with the maximum AUC value of 0.7688.

- The proposed algorithm is comprehensively evaluated in several zero-set settings such as cross-dataset, cross-attack, cross-quality, and in-the-world;

- An extensive comparison with the existing algorithms showcases the effectiveness of the proposed algorithm where the proposed algorithm surpasses them by a significant margin.

## 2 Related Works

In this section, a comprehensive survey of the existing works done towards both the generation of manipulated videos and the defense against them is presented. SWAPPED (Agarwal et al., 2017) is one of the earliest databases prepared using the social media application namely Snapchat. The database contains more than 600 face swap videos and 120 real videos captured in unconstrained environments reflecting real-world applications. Later, (Rossler et al., 2019) prepared one of the largest face manipulation databases covering both identity swap and expression manipulation. Four different attack variants are presented in the database and each type contains 1000 videos along with 1000 real videos. (Dolhansky et al., 2020) released the competition database on DeepFake detection. The database is prepared by Facebook research to enhance the defense against manipulated videos. The database is prepared from more than 3429 paid actors and consists of more than 100,000 face swap videos. (Li et al., 2020c) released the fake videos database of better visual quality that usually surfaces on social media platforms. The database is different from several existing databases which generally consist of visual artifacts helpful in the easy identification of them. The database contains 890 real videos taken from YouTube and high-resolution and color-consistent 5639 Deep-Fake videos. Apart from these sophisticated databases generated using computationally complex systems, (Majumdar et al., 2019) have shown the effect of partial tampering of facial attributes such as blending of an eye and mouth portion on deep face recognition networks. Not only do these facial manipulations utilize pair or more images for manipulation, adversarial noise on face images and presentation attack is also a serious concern (Agarwal et al., 2016; 2021c; 2017b; Goswami et al., 2019; Sanghvi et al., 2020). However, this research focuses on the alteration around DeepFake because of its high impact on society.

Due to the adverse effect of these manipulations, several detection strategies are also proposed ranging from traditional handcrafted features to deep neural networks. (Agarwal et al., 2017) have proposed a feature extraction algorithm effective for highlighting the artifacts that occurred due to the swapping of faces. These DeepFake videos generally contain facial and voice information and utilizing both these visual and audio

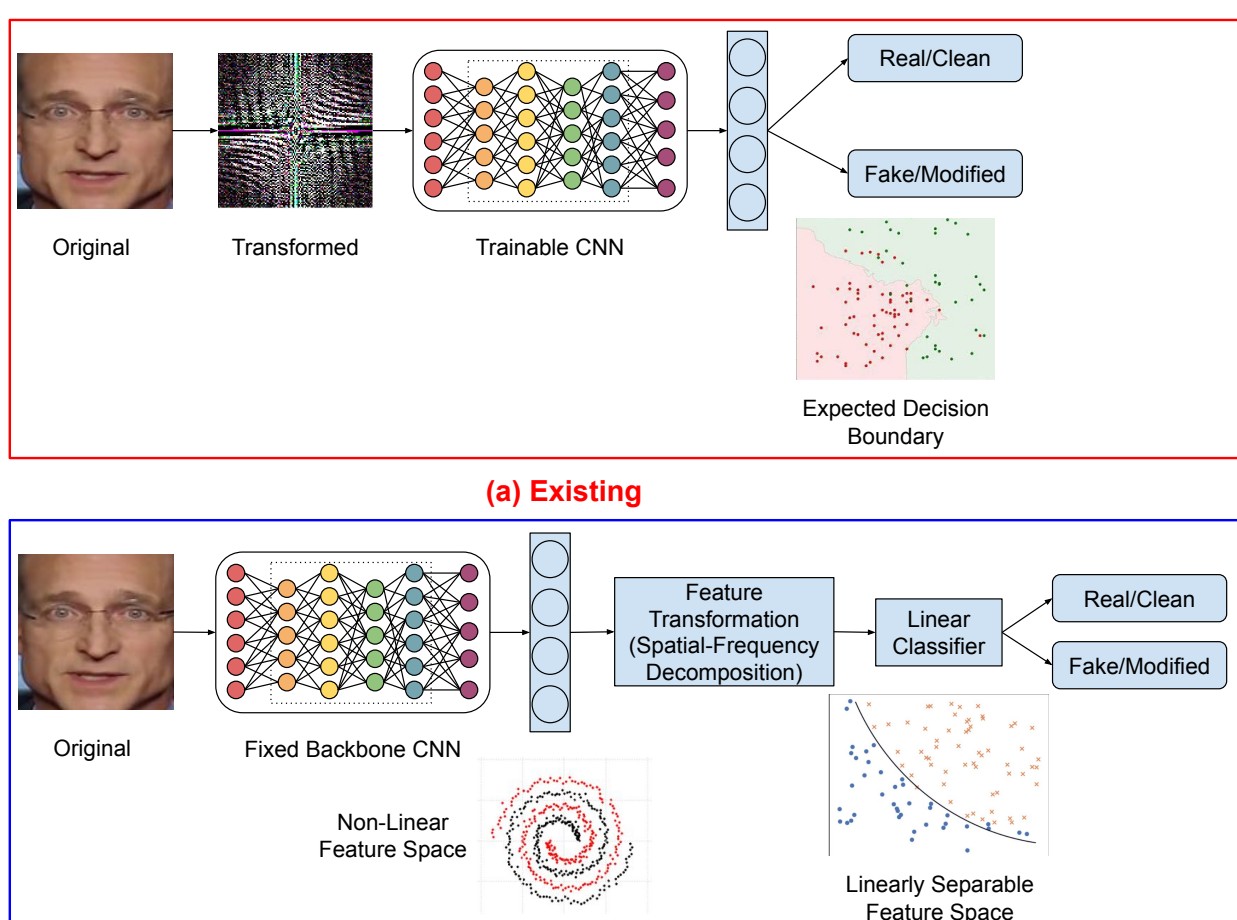

**(a) Existing**

**(b) Proposed**

Figure 3: Comparison between the existing image transformation-based approaches vs. the proposed feature transformation-based algorithm using spatial-frequency decomposition. Input transformations that are applied on either 2D input are computationally complex as compared to the proposed feature transform which works on 1D representation.

features, (Chugh et al., 2020) and (Mittal et al., 2020) have proposed a DeepFake detection algorithm. As discussed earlier, most of the databases exhibit visual artifacts such as head pose and eye blinking, based on these inconsistencies, (Agarwal et al., 2019b) and (Jung et al., 2020) have proposed deepfake detectors. (Rossler et al., 2019) have performed a detailed study with several steganalysis features combined with either traditional classifiers or deep neural networks for manipulation detection. (Wang & Dantcheva, 2020) have proposed a 3DCNN architecture for the identity swap manipulation. To highlight the minute visual artifacts that occurred due to swapping, (Majumdar et al., 2019) have performed high-pass filtering on the input images. The enhanced images combined with the original images are passed through the Siamese type of architecture for the detection of partial swapping of face attributes. (Kumar et al., 2020) have proposed the multi-regional deep neural network architecture for the detection of reenactment manipulation. The authors have reported state-of-the-art results ranging from high-quality videos to tightly compressed videos. (Tolosana et al., 2021) have performed a deepfake detection study by utilizing different components of facial regions. The authors have found that the existing SOTA algorithms are not generalized against recent and complex deepfake datasets. (Fernando et al., 2020) have proposed a memory neural network to counter the deepfake threat. However, as reported through experiments, similar to the existing algorithm, the algorithm is not generalized against unseen attacks. The survey on these face manipulation types and defense against

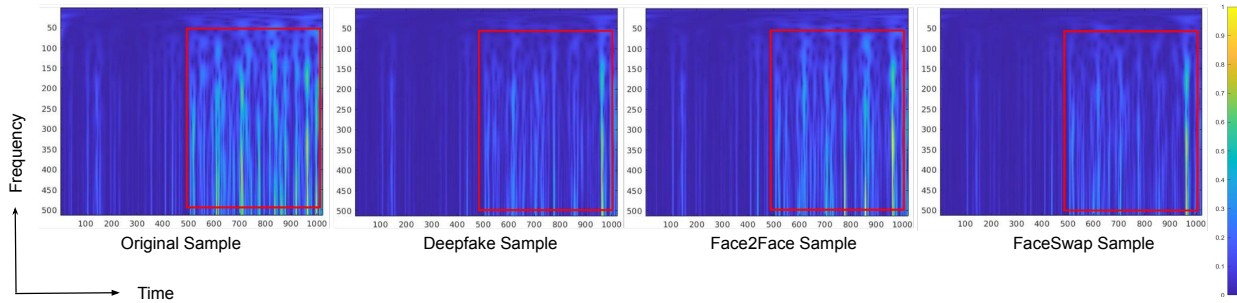

Figure 4: S-transform reflects the potential of differentiating different classes of images including real and several image manipulations. The S-transform heatmap is reported using the images of the FF++ dataset showcases the significant frequency distribution differences between real and fake classes. The red box shows the example of discriminative flow across kinds of data, i.e., real and multiple manipulations. The Y-axis is frequency and the x-axis is time i.e. feature vector from DenseNet. (Best view in color and zoom.)

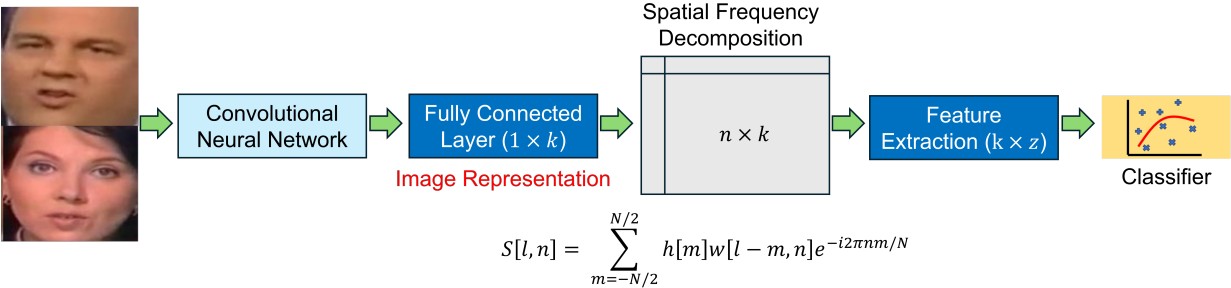

Figure 5: Proposed network for the detection of face-manipulated videos. (Best view in color.)

them can also be found in the survey paper for further study (Mirsky & Lee, 2021; Singh et al., 2020; Tolosana et al., 2020).

From the discussion above, it is observed that the deepfake detection research witnesses a plethora of algorithms; however, their weakness includes the generalization and effectiveness Agarwal & Ratha (2024b) in handling the scenarios outlined in Figure 1. Figure 2 shows the limitations of the recent state-of-the-art algorithms when evaluated in the out-of-distribution samples of unseen attacks or datasets. Another drawback, interestingly with the inception of several high-quality and large-scale datasets, the majority of research is focused on two datasets only namely FF++ (Rossler et al., 2019) and Celeb-DF (Li et al., 2020c). On top of that, only the Celeb-DF dataset is used for cross-dataset generalization. Very little research focuses on cross-attack robustness. Apart from these two generalizations, we assert several other factors are important as reflected in Figure 1. This research dealt with these critical challenges such as experimental evaluation on multiple large datasets and maybe reflecting in-the-wild settings and several evaluation generalization settings to make the algorithm real-world ready.

## 3 Proposed Detection Algorithm to Build Trust on Multimedia Content

The manipulated videos generated using deep learning networks contain significant variations in the high-frequency features of images. By utilizing this information, recent image transformations-based defense approaches have articulated the frequency artifacts (Agarwal et al., 2021c; Chai et al., 2020; Frank et al., 2020; Masi et al., 2020; Qian et al., 2020; Wang et al., 2020) in detecting the fake media. However, these transformations are restricted to the input pixel space which is high dimensional space due to the high resolution of images. Inspired by existing domain knowledge, we have utilized the concept of transformation but at the feature level, to build an efficient deepfake detector. Figure 3 shows the comparison between the proposed and existing transformation-based research. The proposed DeepFake detection algorithm consists

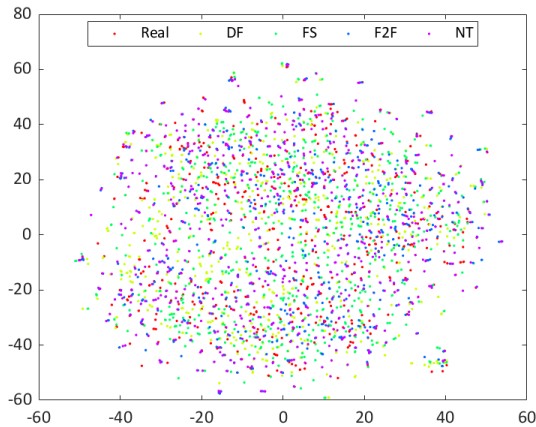 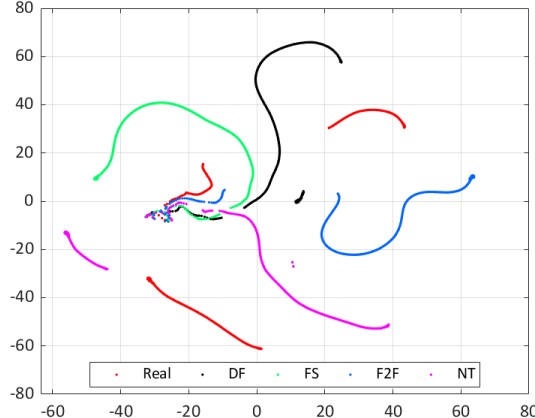

Figure 6: t-SNE (Van der Maaten & Hinton, 2008) visualization of real and different manipulation videos in the original feature space (left) proposed feature transformation space (right). The proposed feature transformation depicts its strength in separating the classes effectively.

of four modules: (i) image representation, (ii) spatial-frequency decomposition to enhance the artifacts, (iii) feature extraction, and (iv) classification.

**Image Representation:** The fully connected feature vector obtained from a convolutional neural network (CNN) is used as an image representation. Since the CNN model used in this research is pre-trained on millions of natural images, we assert that it is highly effective in generating a discriminative and compact representation of an input (Vo et al., 2019; Xie et al., 2016; Ren et al., 2015). In this research, we have used the ImageNet (Deng et al., 2009) pre-trained CNN model for the extraction of a representative vector of an image. It can be defined as follows: $F = CNN(x)$, where, $F$ is a $k$ dimensional 1-D vector representing the input sample($x$).

**Time/Spatial Frequency Decomposition:** Recent studies (Agarwal et al., 2021c; Chai et al., 2020; Frank et al., 2020; Masi et al., 2020; Qian et al., 2020; Wang et al., 2020) have shown effective use of image transformation in the detection of synthetically generated images and deepfake manipulated videos. These studies however mainly utilize the standard frequency transformations such as Fourier or cosine transformations and limit the transformation to the input level only, i.e., only the input images are transformed. These transformed images are then fed into the CNN for end-to-end feature extraction and classification. Fourier transforms (Brigham & Morrow, 1967; Cooley et al., 1969) converts the data from its original spatial space into the different frequency components. In such a case, the original spatial information is lost which might be required for the task at hand. Therefore, a few researchers have used the transformed information as a separate component for the DeepFake examples detection (Majumdar et al., 2019; Masi et al., 2020). These two branch algorithms increase the computational complexity of the entire defense algorithm and are also found not generalizable. In this research, in place of raw pixel space transformation, we have applied the transformation on an image representation using Stockwell Transform (S-transform) (Stockwell et al., 1996). S-transform maps the input to the simultaneous space of spatial/time (in our context, the time axis represents the feature vector from DenseNet) and frequency. The importance of frequency information has been highlighted in the existing synthetic media detection research (Wang et al., 2020; Qian et al., 2020); however, it is only obtained from the input space. The input image space can suffer from several bottlenecks such as the impact of illumination, quality, and compression; while the representation obtained from CNN somewhat mitigates these bottlenecks by learning rich invariant features (Wang & Yeung, 2013; Nanni et al., 2017). Therefore, obtaining the frequency information at the same time retaining the spatial information which is the image representation can help in building a robust detection algorithm. The s-transform helps in achieving that local frequency information (Stockwell, 2007a;b). Apart from that, the S-transform combines the properties of both Fourier transform and wavelet decomposition which are individually found

effective in detecting fake images (Zhang et al., 2019; Wang et al., 2020; Ventosa et al., 2008; Agarwal et al., 2021c). Therefore, decomposing a signal into its corresponding frequency signal along with that spatial (time) information can highlight those individual artifacts that are helpful for the discrimination against fake videos.

In the proposed research, the spatial/time axis is the $k$ dimensional representation of the input image computed from the CNN. The (time)spatial-frequency decomposition through S-transform generates the complex 2D matrix of size $k \times z$ from the 1D representation of an image, where $k$ is the size of the image representation vector and $z$ is the frequency component computed over each value. From the complex values of spatial-frequency decomposition, we have computed the absolute value for further feature extraction and classification.

**Feature Extraction:** Figure 4 shows the magnitude spectrum of the S-transform computed from the real and various types of manipulation videos. Each manipulation shows a significant difference in the frequency spectrum (y-axis) across the time-axis (feature vector) from the real videos. Such significant differences in the manipulation cover a range of techniques such as identity swap (FaceSwap (Kowalski, 2018) and DeepFake (deepfakes, 2018)) and expression swap (Face2Face (Thies et al., 2016)) in the spatial-frequency decomposition makes them perfect for their detection. Therefore, based on such discriminative features of the magnitude values of the spatial-frequency decomposition, we have utilized that for feature extraction. The wavelet energy features at three levels are computed along the frequency axis which acts as the summation of the frequency information at an interval. It leads to a matrix of dimension $k \times 3$, where $k$ is the dimensionality of the image representation and 3 represents the number of energy features. As seen from Figure 4 some frequency stamps are highly discriminative as compared to others. Therefore, we assert that preserving those spectra can provide better classification performance and further reduce the computational cost. To find the discriminative directions best suitable for classification, principal component analysis (PCA) (Pearson, 1901; Wold et al., 1987) has been applied to the extracted features from the training set. For the computation of PCA, the feature matrix of size $k \times 3$ is first converted into the vector of size $1 \times (k \times 3)$. Additionally, Figure 6 also depicts the separation capability of the proposed transformation applied to the feature space.

**Classification:** Once the features are extracted and only the essential components are preserved, a linear support vector machine (SVM) (Cortes & Vapnik, 1995) classifier is trained for the detection of manipulated images. The parameters of the classifier are optimized using a grid search on the training or validation set. We assert that the proposed features transformed the images into space where they are linearly separable and restrict the need for complex non-linear classifiers. Therefore, the linear binary classifier is trained to classify the images into either real or fake classes. Figure 5 shows the overall structure of the proposed algorithm containing multiple steps covering image representation to classification.

## 3.1 Implementation Details

We want to *highlight* that the proposed face manipulation detection algorithm is majorly a 'parameter-free'[1] architecture. We have used the ImageNet pre-trained DenseNet (Huang et al., 2017) CNN network to extract the representation of the input images. The feature representation is of the dimensionality $1 \times 1024$. The representation is then converted into the (time)spatial-frequency decomposition. The decomposition used in this research does not yield any parameter and the default parameter of the transform has been applied (Dash, 2021). The transformation yields the complex-valued matrix of dimension $1024 \times 513$ ($k \times z$), where 513 is the frequency dimension (which is equal to $(k/2) + 1$) and 1024 is the image representation dimension. Later, the magnitude of the spatial-frequency matrix is computed for wavelet energy feature extraction. Again the features extraction step is parameter-free. The only parameters lie in the binary classifier; however, they also do not contain significant parameters.

The proposed **'parameter-free'** architecture makes it a suitable choice for its real-world deployment for the cost-effective device and advanced computing devices. Face manipulation is not only affecting the celebrity class of society but can be harmful to any common person as well. Therefore, the availability of an algorithm that everyone in society can use without worrying about the heavy computation resources can be impactful.

---

[1]It represents that the trainable parameters in the proposed architecture are lying with the classifier only. The CNN has the parameters, although they are fixed in our setting.

The proposed architecture is *'energy-efficient* which should also not be ignored because of the high impact of ML algorithms on the environment (Lu, 2019).

## 4 Experimental Databases

In this research, we have used two popular face manipulation databases namely FaceForensics++ (FF++) (Rossler et al., 2019), Celeb-DF (Li et al., 2020c), DFinal (Ciftci et al., 2020), and deepfake in-the-wild (DFW) (Zi et al., 2020). FF++ is one of the most popular databases because it covers a large spectrum of manipulation concerning identity swaps and expression swaps. We would also like to mention that the videos in the database contain three variations: (i) Raw (original quality), ii) HQ (quantization parameter equal to 23 and referred to as *C-23*), and iii) LQ (quantization parameter equal to 40 and referred to as *C-40*). The compression variations are generated to simulate the real-world processing generally performed on social media uploads. In total, each variant contains 5000 videos, out of which 1000 belongs to each class, i.e., real and four manipulation types. To compute the image representation, faces are first cropped from the input images/videos and resized to a fixed size of $224 \times 224 \times 3$ and are normalized based on the eye coordinates. Apart from this standard processing of images, no other pre-processing has been applied to the face images. Since the number of frames in the existing deepfake videos is very high and they do not contain discriminating information at every frame, therefore, we have randomly selected 10 frames per video for detection. To classify, a video as deepfake or real, the average score of all the frames of the video is used as a score of that video. Such reduction of the frame is inspired by the current literature (Agarwal & Ratha, 2024a) and also helps in reducing the computational cost while maintaining a significant deepfake detection performance.

Celeb-DF and DFinal overcome the limitation of several existing low-quality DeepFake databases by generating the high-quality version which was released over several social media platforms (CtrlShiftFace, 2020). The one probable drawback of the above datasets is that they might not truly reflect the real-world case study of deepfake detection. With this intuition and better support for real-world deepfake detection research, (Zi et al., 2020) have collected a deepfake in-the-wild dataset. The experiments on each dataset are performed using the pre-defined protocol mentioned by the authors of the original paper. For example, the FF++ database comes with a pre-defined training, validation, and testing set and we have used these pre-defined settings for the experimentations and fair comparisons. The results in the paper are reported on the dataset containing both real and attack classes, not just one class.

## 5 Experimental Results and Analysis

In this research, we have conducted the most rigorous experiments and utilized every potential aspect of the databases. To perform the experiments and report the results, the videos in both the databases are preprocessed such as face regions are cropped and normalized to the fixed resolution of dimension $128 \times 128 \times 3$. The performance is reported in terms of video-based classification: where an entire video is classified as real or manipulated.

### 5.1 Seen Attack: Traditional Settings

When the algorithms are trained on each specific manipulation set and tested on that particular manipulation set, the results of the proposed algorithm along with existing algorithms are reported in Table 1. The existing algorithms used by (Rossler et al., 2019) range from hand-crafted features with a traditional classifier to deep neural networks. From the results, it is clear that the steganalysis features yield comparable or better performance than several deep neural networks on 'raw' quality data. Another advantage of such an algorithm is its computational efficiency; however, the algorithm suffers severely on compressed quality images and the detection performance degrades at least 17% and 23% on light-compressed and heavily compressed videos, respectively. The performance of the fine-tuned XceptionNet is the best among all the existing algorithms used in the paper (Rossler et al., 2019). The architecture yields approximately perfect detection performance on 'raw' quality videos. While on mild compression, the XceptionNet is found robust; the architecture shows a significant drop on heavily compressed videos. Another critical disadvantage of the architecture is the

Table 1: Manipulation-specific detection Accuracy (%) of the existing and proposed algorithm. The results are reported on raw and compressed datasets of all four manipulation methods (DF: DeepFakes, F2F: Face2Face, FS: FaceSwap, and NT: NeuralTextures). The accuracy on each subset also includes the accuracy on the pristine (real) videos, i.e., the reported accuracy is the average performance on both the clean and attack images/videos.

| Algorithm | Raw | | | | C-23 | | | | C-40 | | | |
|---|---|---|---|---|---|---|---|---|---|---|---|---|
| | DF | F2F | FS | NT | DF | F2F | FS | NT | DF | F2F | FS | NT |
| (Fridrich & Kodovsky, 2012) | 99.03 | 99.13 | 98.27 | 99.88 | 77.12 | 74.68 | 79.51 | 76.94 | 65.58 | 57.55 | 60.58 | 60.69 |
| (Cozzolino et al., 2017) | 98.83 | 98.56 | 98.89 | 99.88 | 81.78 | 85.32 | 85.69 | 80.60 | 68.26 | 59.38 | 62.08 | 62.42 |
| (Bayar & Stamm, 2016) | 99.28 | 98.79 | 98.98 | 98.78 | 90.18 | 94.93 | 93.14 | 86.04 | 80.95 | 77.30 | 76.83 | 72.38 |
| (Rahmouni et al., 2017) | 98.03 | 98.96 | 98.94 | 96.06 | 82.16 | 93.48 | 92.51 | 75.18 | 73.25 | 62.33 | 67.08 | 62.59 |
| (Afchar et al., 2018) | 98.41 | 97.96 | 96.07 | 97.05 | 95.26 | 95.84 | 93.43 | 85.96 | 89.52 | 84.44 | 83.56 | 75.74 |
| (Chollet, 2017) | 99.59 | 99.61 | 99.14 | 99.36 | 98.85 | 98.36 | 98.23 | 94.50 | 94.28 | 91.56 | 93.70 | 82.11 |
| (Liu et al., 2021b) | – | – | – | – | 97.45 | 98.33 | 97.20 | 90.84 | – | – | – | – |
| (Wu et al., 2020) | – | – | – | – | – | – | – | – | 95.33 | 90.48 | 94.09 | – |
| (Liu et al., 2021a) | – | – | – | – | – | – | – | – | 93.48 | 86.02 | 92.26 | 76.78 |
| (Qian et al., 2020) | – | – | – | – | – | – | – | – | 97.97 | 95.32 | 96.53 | 83.32 |
| (Zhao et al., 2021b) | – | – | – | – | – | – | – | – | 99.73 | 96.38 | 98.20 | 91.79 |
| (Tan et al., 2022) | – | – | – | – | 98.10 | – | – | – | 91.43 | – | – | – |
| (Agarwal et al., 2021a) | – | – | – | – | 98.82 | 99.19 | 99.10 | 94.55 | 97.34 | 93.57 | 94.64 | 81.78 |
| (Pang et al., 2023) | – | – | – | – | 99.64 | 99.64 | 99.28 | 97.50 | 98.21 | 97.50 | 96.07 | 89.64 |
| (Zhao et al., 2023) | – | – | – | – | 99.60 | 99.60 | 100.00 | 96.80 | 98.90 | 96.10 | 97.50 | 92.10 |
| (She et al., 2024) | 99.87 | 99.16 | 98.91 | 90.84 | – | – | – | – | – | – | – | – |
| **Proposed** | **99.90** | **99.99** | **99.97** | **99.98** | **99.97** | **99.95** | **99.15** | **99.50** | **99.90** | **99.65** | **99.00** | **98.70** |

robustness against a variety of manipulation types. For example, the architecture was found robust against 'identity swap' manipulation, i.e., DF and FS, but yields significantly lower performance on 'expression swap' manipulation especially 'NeuralTexture'. As shown in Figure 4 and Figure 6, the proposed algorithm depicts clear differences among the different data types including real and manipulated, which gets reflected in the detection performance. The proposed algorithm shows either **'near-perfect'** or state-of-the-art (SOTA) detection accuracy on each manipulation type and quality type. Such universal detection performance which is agnostic to manipulation types and their quality makes the proposed approach an ideal choice for its real-world deployment.

## 5.2 Cross Attack and Data-Quality

To fully utilize the potential of the FF++ database which is missing from the existing literature, we have performed an evaluation study where one type of manipulation has been used in the training and other manipulations are used for testing. In these experiments, only one *variable* (manipulation type) has changed while another variable (data quality) remains fixed. In other words, when training the detector on the 'raw' quality of one manipulation it is only tested against the 'raw' quality data of other manipulations. As mentioned earlier this type of study is missing from the literature, therefore, we have run the experiments using the best-performing architecture, i.e., XceptionNet, and finetuned it for cross-attack evaluation.

The XceptionNet architecture which shows high detection performance on seen attack images/videos, failed significantly when the attack types were not seen at the time of training. Interestingly we have seen that when the XceptionNet has seen the attack images at the time of training it yields at least 99%, 94.5%, and 82.11% detection accuracy on 'raw', C-23, and C-40 quality images, respectively. Whereas, when the network is tested against unseen attack images where at a time one attack is used for training, the XceptionNet yields only 53.06%, 50.24%, 52.08% average detection accuracy on 'raw', C-23, and C-40 quality images, respectively (Majumdar et al., 2021). A similar low performance against unseen attacks is observed by Cao and Gong (Cao & Gong, 2021). We have also made a comparison with the recently reported results by (Liu et al., 2021b). When the MesoNet, XceptionNet, LAE (Du et al., 2020) models are trained on the F2F set and tested on the FS attack, they yield 47.32%, 49.94%, 63.15%, respectively. The ADD method proposed by (Liu et al., 2021b) also shows a severe drop in the detection performance and yields only 67.02% accuracy. The huge performance drops show the limitation of the existing state-of-the-art network and demand an effective attack agnostic detection algorithm. The proposed research fulfills that gap by producing an attack

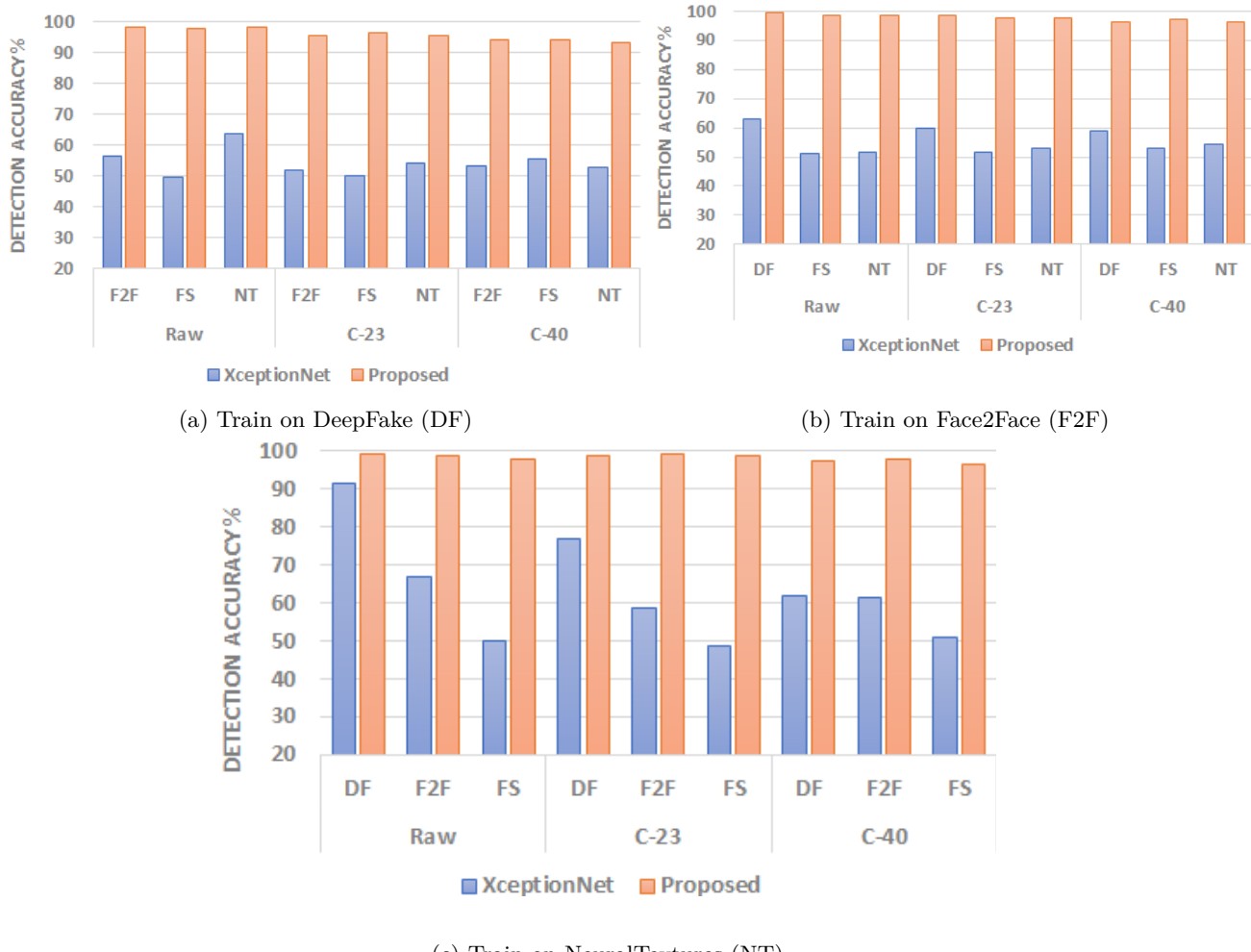

(a) Train on DeepFake (DF)          (b) Train on Face2Face (F2F)

(c) Train on NeuralTextures (NT)

Figure 7: **Attack agnostic** classification accuracy (%) of the best FF++ model, i.e., XceptionNet and proposed algorithm. The detection algorithms are trained on a specific type of manipulation and evaluated on others in the FF++ database. (Best view in color.)

and data quality agnostic algorithm. The proposed algorithm is not only effective in the case of a seen attack and yields the SOTA detection rate but performs similarly even under unseen attack settings as well. The results of this finding are shown in Figure 7.

Figure 8 shows the comparison of the proposed algorithm with two recently proposed generalized deepfake detection algorithms namely Multi-Att (Zhao et al., 2021a), Face X-ray (Li et al., 2020b), Freq-SCL (Li et al., 2021), HFF (Luo et al., 2021), and RECCE (Cao et al., 2022). When the existing algorithms are trained on the DF subset and tested on the remaining, they are found highly ineffective as compared to when they are tested on the DF subset. Whereas, the proposed algorithm yields an attack-agnostic nature and yields significantly better performance than the existing algorithms. In an interesting observation, when the existing algorithms are tested on identity manipulation attacks such as DF and FS, their performance degrades significantly under unseen attack training-testing settings. In contrast, on the expression evaluation manipulations such as F2F and NT, their performance is significantly higher. In both cases, the proposed algorithm yields higher AUC and establishes its attack agnostic strength.

If the proposed algorithm encounters both variables, i.e., attack types and data quality, it is found 'duly' agnostic and yields more than 96% detection performance as reported in Table 2. The prime advantage of such dual robustness is that in the real world, an attacker can come up with a new attack and perform some

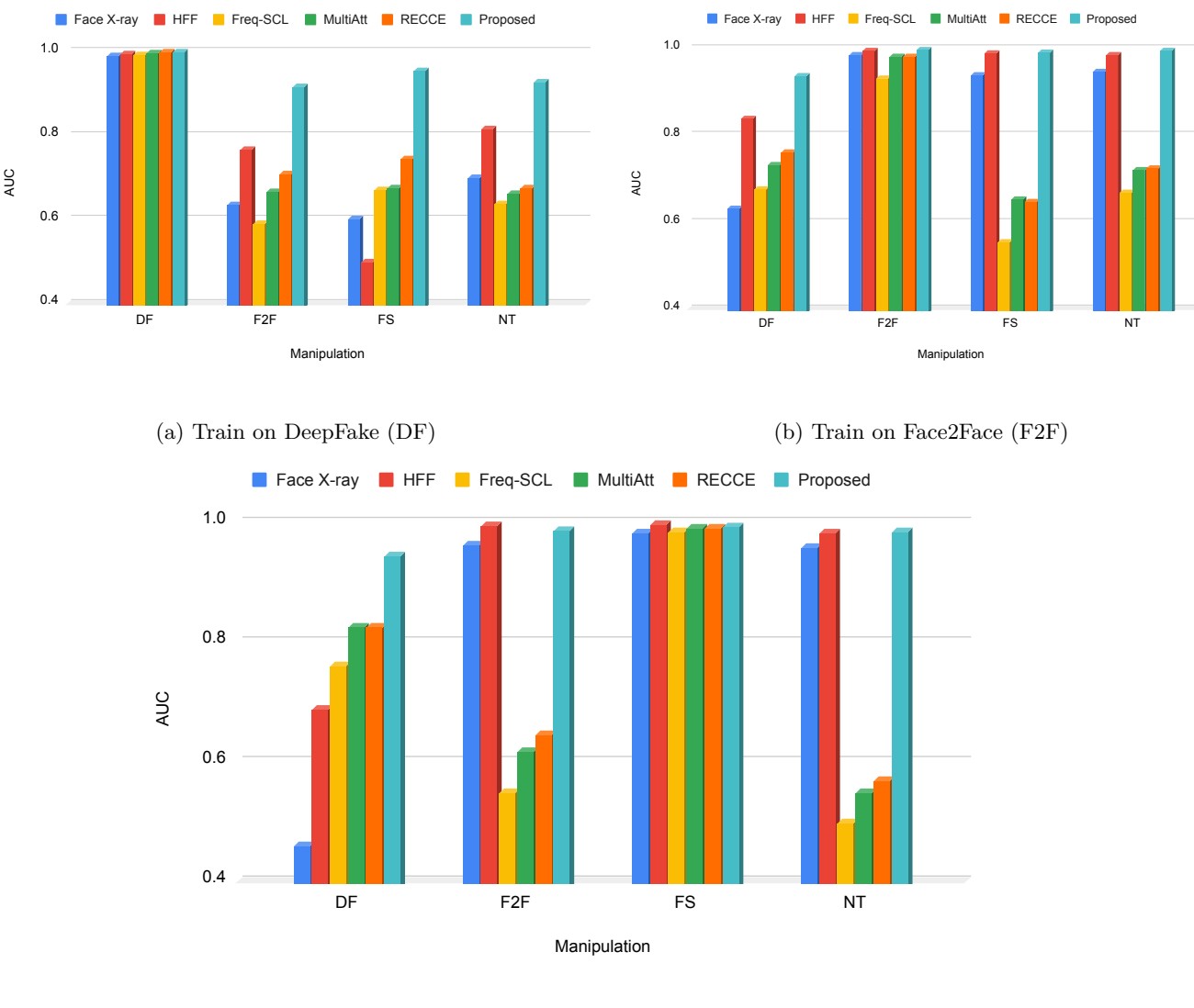

(a) Train on DeepFake (DF)

(b) Train on Face2Face (F2F)

(c) Train on Face swap (FS)

Figure 8: Cross attack evaluation and comparison with the recent state-of-the-art algorithms. The existing algorithms are: Multi-Att (Zhao et al., 2021a), Face X-ray (Li et al., 2020b), Freq-SCL (Li et al., 2021), HFF (Luo et al., 2021), and RECCE (Cao et al., 2022).

different degradation that might not be seen at the time of training. Hence, the detection algorithm must tackle all such unwanted testing conditions that are still missing in the literature and future research must tackle all such extensive evaluation settings. The accuracy of individual experiments about cross quality and attack are reported in Figure 9 showcasing that the proposed algorithm is unbiased in handling different attacks and data quality.

## 5.3   Cross Database Evaluation

(Li et al., 2020c) have performed a comparison of the detection performance of multiple databases using multiple existing DeepFake detection algorithms. The authors claim that the detection of DeepFake images of the Celeb-DF database is most challenging as compared to several other counterpart databases including DFDC (Dolhansky et al., 2019) and DFD (Dufour & Gully, 2020). Therefore, to further showcase the strength of the proposed algorithm, we have utilized the Celeb-DF database, and the database is used for evaluation

Table 2: Attack and quality agnostic performance of the proposed algorithm on the FF++ database. The proposed algorithm is duly agnostic against both the variables in the database, i.e., attack types and image quality.

| Training | | Agnostic Variables | | Average |
|---|---|---|---|---|
| Attack | Quality | Attack | Quality | Accuracy |
| DF | Raw | FS, F2F, and NT | C-23 and C-40 | |
| | C-23 | | Raw and C-40 | |
| | C-40 | | Raw and C-23 | |
| FS | Raw | DF, F2F, and NT | C-23 and C-40 | |
| | C-23 | | Raw and C-40 | |
| | C-40 | | Raw and C-23 | **98.61 ±** |
| F2F | Raw | DF, FS, and NT | C-23 and C-40 | **0.43** |
| | C-23 | | Raw and C-40 | |
| | C-40 | | Raw and C-23 | |
| NT | Raw | DF, FS, and F2F | C-23 and C-40 | |
| | C-23 | | Raw and C-40 | |
| | C-40 | | Raw and C-23 | |

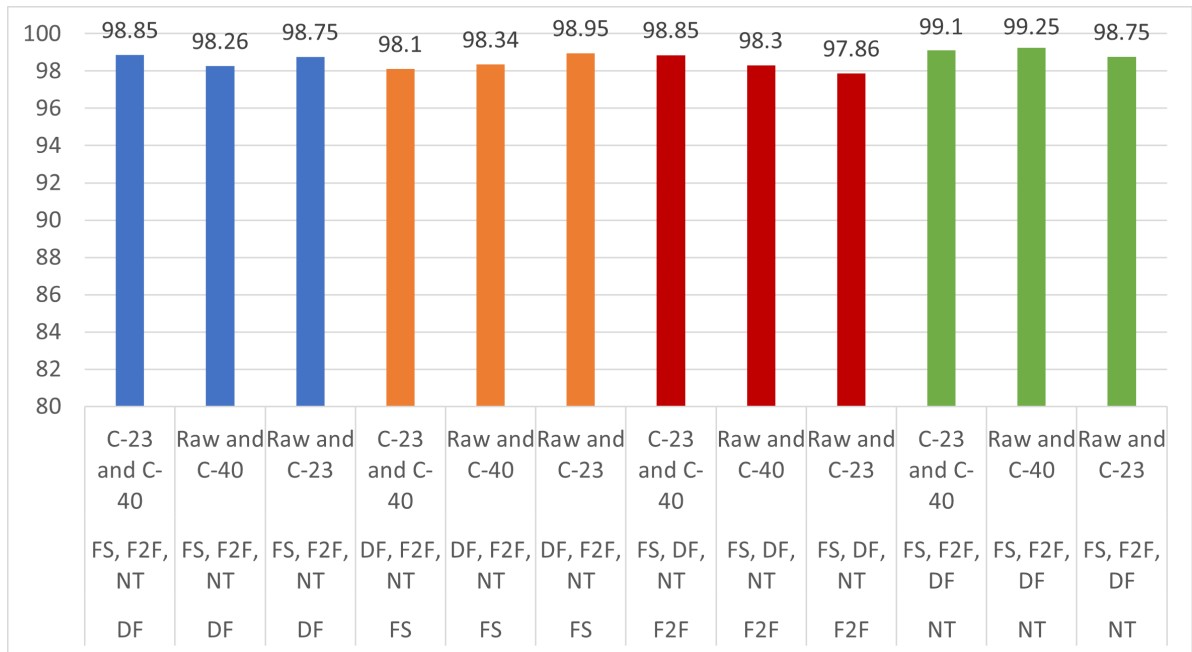

Figure 9: Attack and quality agnostic performance of the proposed algorithm on the FF++ database. For example, the proposed algorithm is trained on DF (first three bars) and tested on remaining attacks of different testing image qualities (leftmost bar: trained on Raw quality and tested on unseen image qualities C23 and C40). The lower variation among the accuracies showcases that the proposed is robust and not biased towards any particular attack or data quality.

under cross-database settings only. Each attack and quality image of FF++ databases is used one at a time for training and evaluated on the Celeb-DF database. The FF++ database contains twelve variants, i.e., four attacks and three data qualities. The results reported in Table 3 show the average performance of all twelve variants, where each variant yields at least 98% detection AUC performance on the Celeb-DF database. Along with AUC, the proposed algorithm archives 92.76% real image detection accuracy and 96.78% deepfake image detection accuracy on the Celeb-DF dataset. The average detection accuracy is 94.77%. The performance of the proposed algorithm is compared against several existing state-of-the-art algorithms namely (i) Two-stream (Zhou et al., 2017), (ii) Meso4 (Afchar et al., 2018), (iii) MesoInception4 (Afchar et al., 2018), (iv)

Table 3: Cross-dataset evaluation (AUC) and comparison with complex state-of-the-art algorithms on Celeb-DF (Li et al., 2020c). The results on FF++ are reported under the same database training-testing and most of the algorithms achieve almost perfect performance. However, each algorithm suffers a drastic drop in performance when an unseen database comes except the proposed algorithm.

| Method | FF++ | Celeb-DF |
|---|---|---|
| MesoInception4 (Afchar et al., 2018) | 0.83 | 0.53 |
| Xception-raw (Rossler et al., 2019) | 0.99 | 0.48 |
| Xception-c23 (Rossler et al., 2019) | 0.99 | 0.65 |
| Xception-c40 (Rossler et al., 2019) | 0.95 | 0.65 |
| Multi-task (Nguyen et al., 2019a) | 0.76 | 0.54 |
| Capsule (Nguyen et al., 2019b) | 0.96 | 0.57 |
| DSP-FWA (Li & Lyu, 2019) | 0.93 | 0.64 |
| Face-XRay (Li et al., 2020b) | 0.99 | 0.74 |
| $F^3$-Net (Qian et al., 2020) | 0.97 | 0.65 |
| EfficientNet-B4 (Tan & Le, 2019) | 0.99 | 0.64 |
| MD-CSDNetwork (Agarwal et al., 2021a) | 0.99 | 0.68 |
| (Nirkin et al., 2021) | 0.99 | 0.66 |
| ProtoPNet (Chen et al., 2018) | 0.98 | 0.69 |
| DPNet (Trinh et al., 2021) | 0.99 | 0.68 |
| DPNet - c40 (Trinh et al., 2021) | 0.90 | 0.72 |
| M2TR (Wang et al., 2021) | 0.99 | 0.66 |
| Multi-Attention (Zhao et al., 2021a) | 0.99 | 0.67 |
| MFFNet (Zhao et al., 2021b) | 0.99 | 0.75 |
| ADD (Liu et al., 2021b) | 0.97 | 0.66 |
| STIL (Gu et al., 2021) | 0.97 | 0.75 |
| PCL + I2G (Zhao et al., 2021c) | 0.99 | 0.81 |
| CORE (Ni et al., 2022) | 0.99 | 0.79 |
| RECCE (Cao et al., 2022) | 0.99 | 0.69 |
| UIA-ViT (Zhuang et al., 2022) | 0.99 | 0.82 |
| SCL-KD (Lin et al., 2022) | 0.99 | 0.69 |
| Trans-FCA (Tan et al., 2022) | 0.99 | 0.78 |
| Chen et al. (2022) | 0.98 | 0.80 |
| Forensics Symmetry (Li et al., 2023a) | 0.99 | 0.58 |
| ISTVT (Zhao et al., 2023) | 0.99 | 0.84 |
| (She et al., 2024) | 0.99 | 0.92 |
| (Yan et al., 2024) | − | 0.83 |
| Khormali & Yuan (2024) | 0.99 | 0.88 |
| **Proposed**[*] | **0.99** | **0.98** |

[*]The proposed algorithm yields more than 94.77% detection accuracy.

HeadPose (Yang et al., 2019), (v) FWA (Yang et al., 2019), (vi) VA-MLP (Matern et al., 2019), (vii) VA-LogReg (Matern et al., 2019), (viii) Xception-raw (Rossler et al., 2019), (ix) Xception-C23 (Rossler et al., 2019), (x) Xception-C40 (Rossler et al., 2019), (xi) Multi-task (Nguyen et al., 2019a), (xii) Capsule (Nguyen et al., 2019b), (xiii) DSP-FWA which is an improved variant of FWA. Apart from these algorithms, the comparison with recent complex algorithms is also performed including MD-CSDNetwork (Agarwal et al., 2021a), (Nirkin et al., 2021), ProtoPNet (Chen et al., 2018) DPNet (Trinh et al., 2021), DPNet - c40 (Trinh et al., 2021), M2TR (Wang et al., 2021), Multi-Attention (Zhao et al., 2021a), MFFNet (Zhao et al., 2021b), ADD (Liu et al., 2021b), STIL (Gu et al., 2021), PCL + I2G (Zhao et al., 2021c), and self-supervised learning (SSL) techniques Khormali & Yuan (2024); Chen et al. (2022). Another desired property of a strong defense is the effectiveness against unseen databases and the performance of the proposed algorithm in that agnostic direction further claims that it is the best possible option for manipulation detection.

Table 4: The results on the individual quality images of the Celeb-DF database show that the proposed algorithm is 'robust' against image quality.

| Algorithm | Original | C-23 | C-40 |
|---|---|---|---|
| FWA | 56.9 | 54.6 | 52.2 |
| Xception-C23 | 65.3 | 65.5 | 52.5 |
| Xception-C40 | 65.5 | 65.4 | 59.4 |
| DSP-FWA | 64.6 | 57.7 | 47.2 |
| Proposed | **99.84** | **98.25** | **97.89** |

Table 5: Attack, database (DB), and quality agnostic performance of the proposed algorithm through the FF++ and Celeb-DF databases. The proposed algorithm is **'tri'** agnostic against each possible variable in the domain, i.e., attack types, database, and image quality.

| Training variables | | | Test Agnostic Variables | | | Average |
|---|---|---|---|---|---|---|
| DB | Attack | Quality | DB | Attack | Quality | Accuracy |
| FF++ | DF | Raw | Celeb-DF | DF | C-23 and C-40 | **98.46 ± 1.24** |
| | | C-23 | | | Raw and C-40 | |
| | | C-40 | | | Raw and C-23 | |
| | FS | Raw | | | C-23 and C-40 | |
| | | C-23 | | | Raw and C-40 | |
| | | C-40 | | | Raw and C-23 | |
| | F2F | Raw | | | C-23 and C-40 | |
| | | C-23 | | | Raw and C-40 | |
| | | C-40 | | | Raw and C-23 | |
| | NT | Raw | | | C-23 and C-40 | |
| | | C-23 | | | Raw and C-40 | |
| | | C-40 | | | Raw and C-23 | |
| Celeb-DF | DF | Raw | FF++ | DF, FS, F2F, NT | C-23 and C-40 | |
| | | C-23 | | | Raw and C-40 | |
| | | C-40 | | | Raw and C-23 | |

Similar to the FF++ database, the Celeb-DF also contains the videos in three image qualities: (i) raw, (ii) C-23, and (iii) C-40. It can be seen from Table 4 that the performance of the existing algorithms significantly degrades with the quality of the images/videos. Whereas, the performance of the proposed algorithm remains the same across each data quality. We want to mention that the proposed algorithm is trained only on FF++ (not seen Celeb-DF images) and tested on individual quality images of Celeb-DF.

### 5.4 Cross Quality, Attack, and Database Variations

The strong manipulation detection algorithm must have considered all possible variables while evaluating the performance. The three variables that might be possible in the detection databases are (i) attacks (identity swap or expression swap), (ii) quality (raw or compressed), and (iii) database. We have earlier shown the effectiveness of the proposed algorithm under two variables using the FF++ database and the third using Celeb-DF testing. However, now we have to consider all three parameters simultaneously and evaluate the agnostic nature of the proposed algorithm against them. The finding showcased in Table 5 establishes the desired effect of the proposed algorithm which is missing in the existing algorithms.

### 5.5 In-the-wild Detection

While the previous datasets are popular benchmark datasets in the deepfake detection study, the recent advancement in deepfake research has introduced several high-quality videos that reflect real-world conditions. Henceforth, to study the effectiveness of our proposed algorithm in deepfake detection, we have utilized two

Table 6: DFinal database (Ciftci et al., 2020) results. The proposed algorithm with DenseNet image representation yields state-of-the-art deepfake detection performance on such a challenging high-quality dataset.

| Algorithm | Face ↑ | Video ↑ |
|---|---|---|
| Simple CNN | 54.56 | 48.88 |
| InceptionV3 (Szegedy et al., 2016) | 60.96 | 68.88 |
| Xception (Chollet, 2017) | 56.11 | 75.55 |
| DenseNet (Huang et al., 2017) | 58.71 | 77.05 |
| VGG (Simonyan & Zisserman, 2014) | 52.10 | 70.85 |
| ConvLSTM (Xingjian et al., 2015) | 44.82 | 48.83 |
| V1 (Tariq et al., 2018) | − | 82.22 |
| V3 (Tariq et al., 2018) | − | 73.33 |
| Emsemble (Tariq et al., 2018) | − | 80.00 |
| Fake Catcher (Ciftci et al., 2020) | 87.62 | 91.07 |
| Deepfake Catcher (Agarwal & Ratha, 2024a) | 91.26 | 94.78 |
| **Proposed with Xception** | **92.23** | **94.89** |
| **Proposed with DenseNet** | **94.65** | **97.80** |

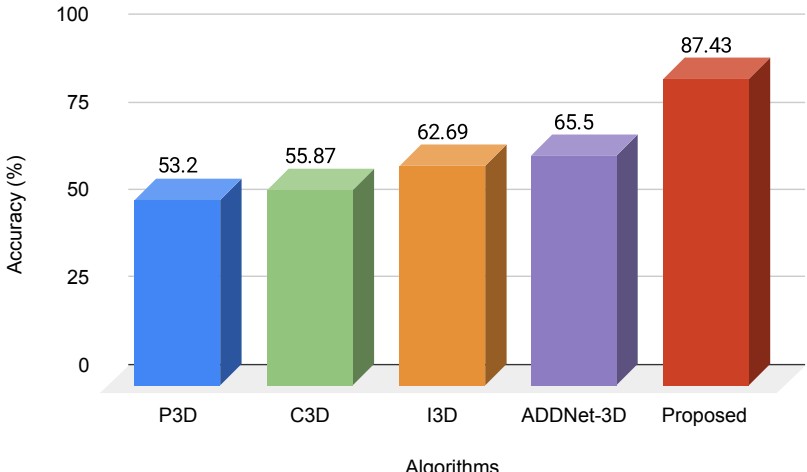

Figure 10: Deepfake detection accuracy (%) on the DFW dataset. Our performance is compared with I3D (Carreira & Zisserman, 2017), P3D (Qiu et al., 2017), C3D (Tran et al., 2014), ADDNet-3D (Zi et al., 2020).

datasets namely DFinal (Ciftci et al., 2020) and DFW (Zi et al., 2020). The DFinal dataset is claimed to be a high-quality deepfake video dataset and is challenging as compared to other datasets. The detection results of the proposed algorithm along with state-of-the-art algorithms are reported in Table 6. The detection results on another challenging dataset namely DFW are also reported in Figure 10. The proposed algorithm surpasses the existing algorithms by at least 21.93%. The effectiveness of the proposed algorithm on such challenging datasets and surpassing several state-of-the-art algorithms reflects its superiority in identifying the deepfake threat. The extensive experimental evaluation helps in establishing our aim of developing a unified deepfake detection algorithm that is agnostic to several challenging dimensions such as dataset, manipulation, and image quality.

## 5.6 Analysis

We have also studied the current strength and bottleneck of the proposed algorithm which can potentially help in building an advanced version of the algorithm and further detecting the deepfake samples. Figure 11 shows some correctly and incorrectly classified samples of both classes. The prime reasons for misclassification

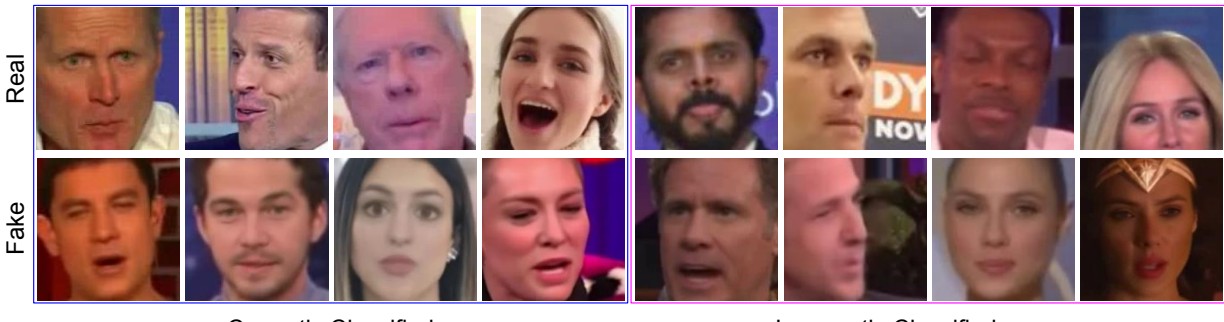

Figure 11: Correct and incorrect prediction samples to study what are the current bottlenecks and strengths of the proposed algorithm. The first row is the real samples and the second row is the fake samples. Interestingly, the majority of the incorrect predictions are somewhat blurry, low quality, and contain a partial face region.

we can see from the images are low quality, blurriness of the images, and low facial region. Although, we want to highlight it is not observed that each image with these characteristics gets misclassified; however, we assert that these might be the potential reasons for incorrect prediction. We want to highlight the interesting strength of the proposed algorithm: it shows good resilience in handling poses, expression, contrast, and quality of the images. On top of that, we have observed there is no bias towards gender modality and the algorithm is fair in detecting both gender images.

We have also analyzed the performance of the proposed algorithm concerning individual classes, i.e., real and attack. The prime reason for such evaluation is that the existing algorithms are found highly biased towards the attack class and yield poor performance of the real class (Nirkin et al., 2021; Rossler et al., 2019). As mentioned earlier the reported results correspond to both real and fake classes. The proposed deepfake detection algorithm is not found biased towards any particular class and yields approximately similar ($\pm 0.56\%$) accuracy in both classes. For example, if the proposed algorithm yields 98.70% NT examples detection accuracy (Table 1), then the detection accuracy on real images is 98.64% and detection accuracy of NT attack is 98.76%.

### 5.7 Discussion

S-transform helps in capturing the local frequency information. Research works show that synthetic media has either suppressed frequency information or has them in excess (Durall et al., 2020). S-transform is effective in detecting these frequency bursts. Frequency information has proven effective in the detection of various synthetic media and adversarial noise detection and is one of the primary differential components between real and fake images (Agarwal et al., 2021a;c; Frank et al., 2020). On top of that, the proposed algorithm also utilizes the wavelet energy features from transformed features. The wavelet energy features contain the information related to both approximation (low frequency) and high-frequency content of an image) (Akbarizadeh, 2012). Figure 4 also shows that the energy information in the real images is highly preserved and suppressed in the fake images of different types. Therefore, The proposed combination of S-transformation which attenuates the frequency artifacts, and acquisition of these artifacts in the form of energy features through wavelet decomposition pave the way for an effective deepfake detection algorithm. The extensive experimental evaluation backs our understanding and yields state-of-the-art and generalized deepfake detection performance. However, as shown in Figure 11, the proposed algorithm is not perfect and we should not be confused that the field of deepfake detection is entirely solved. We believe further advancement is needed to tackle several real-world variations that might be exhibited in future advanced deepfake videos.

## 5.8 Low Computational Cost

The proposed research aims to build a robust solution and provide it at a lower computational cost. As mentioned earlier, the steps involved in the proposed approach are parameter-free, excluding the SVM classifier's parameter optimization. We want to highlight that the linear SVM classifiers are also not high enough to yield significant computational costs. For example, on the T4 GPU of a Google Colab platform, the DenseNet feature extraction took 10 seconds on entire test videos of FF++ (140 videos or 1400 frames of size $224 \times 224 \times 3$). Whereas, the s-transform, dimensionality reduction, and classification take 15 seconds. We want to highlight that except for DenseNet feature extraction, everything else has been on a CPU platform without parallel programming using Matlab.

## 6 Conclusion and Impact

Manipulated images/videos including DeepFakes have created havoc among society, research personnel, and various governments. The manipulation can be broadly classified into two groups: (i) identity swap: where an entire face of a person is swapped with the face of another person, and (ii) expression swap: where only certain part(s) of a face are manipulated for the desirable effects. By looking at the seriousness of the issue, in this research, we have proposed a state-of-the-art detection algorithm based on simultaneous spatial-frequency signal transformation. The algorithm is evaluated on two large-scale databases that constitute various manipulation types. The experiments are conducted on several challenging scenarios which are still missing from the literature on defense. The proposed algorithm is found state-of-the-art reaching an almost perfect rate at each desirable real-world condition including unseen attack, unseen image degradation due to compression, and unseen database. Another significant advantage of the proposed algorithm is its almost *'parameter-free'* nature and hence does not need heavy computational resources and saves energy significantly. The said advantages make it a viable option for anyone around the globe to use without worrying about computational requirements.

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
