# OpenReview forum: "Fast and Generalized DeepFake Detector Through Feature Space Transformation"
_TMLR — Rejected by TMLR_

### Review · Reviewer_FXAY · 2024-12-18

**Summary Of Contributions:**

This work studies the problem of distinguishing deep fake-generated face videos from natural face images. The key idea is apply a space-frequency transform onto image neural features and classify the transformed feature through SVM. The proposed method contains three steps. It first extract feature from images using a ImageNet pretrained neural network backbone. It then apply a space-frequency transform to the feature. Finally the transformed features are fed to an SVM classifier to predict whether the image is generated or not. Experimental results show that the proposed method achieves superior accuracy over compared existing methods, and it is more robust to compression artifacts.

**Audience:**

No

**Broader Impact Concerns:**

I have no ethical concern.

**Claims And Evidence:**

No

**Requested Changes:**

1. Please revise citation format and avoid styles like "Agarwal et al. Agarwal et al. (2017a)".
2. In the sentence "A novel DeepFake detection algorithm is presented by utilizing an amalgamation of various machine
learning eras to identify the discriminating cues helpful for classification", it is very unclear what "an amalgamation of various machine
learning eras" means. Please rephrase.
3. The second and the third bullet point in the contributions is not clear and the languages are ambiguous. Please rewrite.
4. "As discussed earlier, most of the databases exhibit visual artifacts such as head pose and eye blinking, based on these inconsistencies", I don't understand why "head pose and eye blinking" are "artifacts". Please rephrase. Besides, head pose and eye blinking are not parallel forms and the terms should be changed.
5. What does "minute visual artifact" mean? Please revise.
6. For Figure 4, please add color bars and axes titles. "high amount of energy information" is not clear. Please rephrase.
7. "As discussed, several research algorithms are presented in the literature however there are several critical reasons which make the proposed study impactful." Please rewrite.
8. "The manipulated videos are generated using artificial operations and lack natural frequency and spatial compactness." "Natural frequency" is not a well-known term and should be defined. "Spatial compactness" is also not defined and unclear. Please rewrite.
9. "The idea of utilizing the CNN is its eﬀectiveness in handling the variation such as translation, and minute visual artifacts due to environmental factors such as natural noises in the images." Please rewrite. Please also consider rewriting the entire paragraph since it is not very informative.
10. "have shown eﬀective use of transformation in the detection of synthetically". The term "transformation" itself is not clearly defined. Do you mean time-frequency transforms?
11. "For that for the first time in the deepfake detection research, we have utilized the Stockwell Transform (S-transform) Stockwell et al. (1996) which maps the input to the simultaneous space of spatial and frequency." Please revise the language.
12. "Apart from that, the S-transform combines the properties of both Fourier transform and wavelet decomposition which we believe helps in learning robust features". "We believe" is not going to be strong enough for a scientific finding. It either helps or not. Please rewrite to be more rigorous.
13. "the spatial axis is the k dimensional representation" What does "spatial axis" mean?
14. The feature fed into the transform is a 1x1024 vector. Do you mean that you treat this as a sequence in time and apply the transform? How do you generate 513x1024 out of it? Please be more specific on this. Besides, the rationale behind this operation is unclear. Please explain.
15.Fig 2, 7, and 9 are blurry and of low quality. Please include high quality figures for readability.
16. I don't see anything interesting as described in the caption in Fig. 11. Please explain why this figure is interesting.
17. An ablation study is missing. Please at least compare the proposed method with simple linear probing. You may refer to papers referenced above as [1,2]

**Strengths And Weaknesses:**

# Strength

The proposed method is conceptually straight-forward and easy to understand / implement. Experimental results show promising overall performance compared to existing methods.

# Weaknesses

1. **The main idea is trivial**. The key idea in the proposed method is simply applying a space-frequency transform over neural feature and use an SVM to do classification. There is no technical difference between this technique and linear probing (i.e. training one linear classifier with pretrained self-supervised backbone), which has been widely used in existing self-supervised learning research ([1, 2] to name some of them). Despite the good performance, the reason for the additional robustness and performance gain is unclear. In other words, this paper neither analyzes nor reveals the knowledge of why the proposed simple algorithm can lead to significant gain.

2. **Claim / Motivation is unclear**. The motivation of low carbon emission is not solid. Even though training the linear classifier is easy and fast, the backbone itself is trained on large datasets and also require a lot of energy. It is unfair and meaningless to compare other methods' training time to the SVM training time. Besides, the proposed method does not show advantage in inference complexity, which is actually the main source of carbon emission if the method is deployed.

3. **Presentation quality needs much improvements**.  The paper is not properly written and hard to read. Some figures are not clear enough. The flow needs further revision and some terms are not clearly defined. Please see the *requested changes* section for detailed comments.


[1] Grill, Jean-Bastien, et al. "Bootstrap your own latent-a new approach to self-supervised learning." Advances in neural information processing systems 33 (2020): 21271-21284.
[2] Chen, Ting, et al. "Big self-supervised models are strong semi-supervised learners." Advances in neural information processing systems 33 (2020): 22243-22255.

---

> ### Author Response · Authors · 2025-01-01
> **Response**
>
> We want to thank the reviewer for providing constructive comments.
>
> 1. Based on the suggestion, we have updated the citation format.
>
> 2-3. We have updated the bullet points mentioned in the contributions including the rephrasing of the first bullet point.
>
> 4. Head pose and eye blinking are artifacts in a sense suppose if two identities that are going to be deepfake have very different poses, then due to misalignment head pose artifact will occur. Similarly, eye blinking in synthetic videos might have a different number of occurrences than the eye blinking pattern of a real video. These artifacts are already been used in the existing research. For example, [F] mentioned: “For a healthy adult human, generally, between each blink is an interval of 2-10 seconds but the actual rates vary by individual, and the length of a typical blink is 0.1-0.4 seconds/blink2. As such, we should expect to observe spontaneous eye blinking from a video of real humans with the aforementioned frequency and duration. However, this is not the case for many DeepFake videos.”
>
> 5. The visual artifacts include inconsistent facial features, jitter, flickering, missing reflections, and missing details in the eye and teeth areas. The researchers have extensively used these inconsistencies to detect deepfake videos [C,D]. The head pose and eye blinking inconsistencies are used in [E] and [F]. For example, the algorithm in [E] is motivated by the following facts: "3D head pose corresponds to the rotation and translation of the world coordinates to the corresponding camera coordinates and due to misalignment errors in landmark locations occurs."
>
> 6. Based on the suggestion of the reviewer, we have added the axes titles and color bar. Further, the caption is also updated. As pointed out by the reviewer U5rS, we have added the axis titles where the time axis represents the feature computed from the CNN.
>
> 7. Updated the sentence as follows: “From the discussion above, it is observed that the deepfake detection research witnesses a plethora of algorithms; however, their weakness includes the generalization and effectiveness in handling the scenarios outlined in Figure 1.”
>
> 8. We have updated the sentence to “The manipulated videos generated using deep learning networks contain significant variations in the high-frequency features of images.”
>
> 9. Thanks for the suggestion, we have updated the entire paragraph.
>
> 10. Thanks for the suggestion, we have updated the sentence. The existing studies do not utilize the transformation of feature vectors but perform frequency transformation of input images.
>
> 11. We have updated the sentence as follows: “In this research, in place of raw pixel space transformation, we have applied the transformation on an image representation using Stockwell Transform (S-transform) (Stockwell et al., 1996). S-transform maps the input to the simultaneous space of spatial/time (in our context, the time axis represents the feature vector from DenseNet) and frequency.”
>
> 12. We have updated the sentence as follows: “Apart from that, the S-transform combines the properties of both Fourier transform and wavelet decomposition which is individually found effective in detecting fake images (Zhang et al., 2019; Wang et al., 2020; Ventosa et al., 2008; Agarwal et al., 2021c).”
>
> 13. The spatial axis here represents the time axis which is the feature representation computed from an image. As correctly pointed out by Reviewer U5rS, i.e. y-axis is the frequency (z) and the x-axis is k (“time”) i.e. feature vector from CNN. We have updated Figure 4 as well based on the suggestion of the reviewer.
>
> 14. We have updated the Figures for better readability. Further, 1024 is the dimensionality of the feature vector provided by the DenseNet model. S-transform provides the frequency components which is half (i.e., 512) of the dimension of the feature dimension (i.e., 1024). The remaining frequency component belongs to the zero (center) frequency row; henceforth, the frequency dimension is 513 in total.
>
> **Implementation:** The implementation of s-transform is based on its original formulation which includes the computation of Fourier transform coupled with Gaussian window. We will release the source for the reproducibility of the S-transformation components. The text has been updated to clarify the dimension obtained after s-transform.
>
> 15. We have updated the Fig. 11 to improve its quality and better reflect the observations made in this Figure. As mentioned, the figure shows the majority of the misclassified samples are of low quality, blurry, and contain partial face region. For example, images in the third and fourth column of misclassified samples.

---

> > ### Author Response · Authors · 2025-01-01
> > **Response-2**
> >
> > 16. Comparison with Self-Supervised Learning: SSL algorithms aim to learn discriminative features from vast quantities of unlabeled instances without relying on human annotations. In this context, we can say, that the proposed algorithm utilizes the pre-trained model and uses it only on the data to extract features without utilizing the labels. However, in our context, we don’t we do not define any pretext task or generate any pseudo-labels for fine-tuning a deep model.
> >
> > However, based on the suggestion of the reviewer, we have now added the comparison with two state-of-the-art deepfake detection algorithms utilizing the concept of self-supervised learning [A,B]. The comparative results are added in Table 3 of the updated paper. Further, under cross-attack settings, the performance of the proposed algorithm is at least 18% better than these existing SSL techniques.
> >
> > 17. Inference Time: We have rephrased the entire paragraph including removing a claim of the low carbon emission. However, it can be observed that the inference time of the proposed algorithm is minimal (close to a second on a single GPU without parallel programming and CPU processing). The prime reason can be seen from the fact the feature vectors are also heavily reduced due to dimensionality reduction using PCA and linear classifier. The dimensionality reduction and classification have been done on the CPU using Matlab. Only the DenseNet feature extraction has been performed on a GPU system.
> >
> > 18. Presentation Quality: We want to thank the reviewer for all the detailed comments. We have carefully proofread the paper to improve its presentation quality and readability.
> >
> > [A] Khormali A, Yuan JS. Self-supervised graph Transformer for deepfake detection. IEEE Access. 2024.
> >
> > [B] Chen L, Zhang Y, Song Y, Liu L, Wang J. Self-supervised learning of adversarial example: Towards good generalizations for deepfake detection. In Proceedings of the IEEE/CVF conference on computer vision and pattern recognition 2022 (pp. 18710-18719).
> >
> > [C] Mirsky Y, Lee W. The creation and detection of deepfakes: A survey. ACM computing surveys (CSUR). 2021 Jan 2;54(1):1-41.
> >
> > [D] Nguyen TT, Nguyen QV, Nguyen DT, Nguyen DT, Huynh-The T, Nahavandi S, Nguyen TT, Pham QV, Nguyen CM. Deep learning for deepfakes creation and detection: A survey. Computer Vision and Image Understanding. 2022 Oct 1;223:103525.
> >
> > [E] Yang X, Li Y, Lyu S. Exposing deepfakes using inconsistent head poses. InICASSP 2019-2019 IEEE International Conference on Acoustics, Speech and Signal Processing (ICASSP) 2019 May 12 (pp. 8261-8265).
> >
> > [F] Li Y, Chang MC, Lyu S. In ictu oculi: Exposing AI created fake videos by detecting eye blinking. In2018 IEEE International workshop on information forensics and security (WIFS) 2018 Dec 11 (pp. 1-7).

---

### Review · Reviewer_U5rS · 2024-12-20

**Summary Of Contributions:**

The described method enhances deepfake detection by applying a frequency decomposition (via the S-transform) not directly on raw images as many contemporary methods do, but on feature representations derived from a pre-trained DenseNet. The authors posit that using feature space rather than image space allows the pre-trained network to compress invariant (and less useful) information about illumination, image composition etc. into “essential components” that preserve the most useful aspects of an image and improve deepfake detection. This produces a matrix that can be thought of as a spatial-frequency decomposition of essential image features. It has been previously noted in several works that these spatial and frequency features can illuminate artifacts caused by artificial interventions of the kind employed by deepfake methods, but featurization via S-transform is novel. Furthermore, the authors claim this greatly enhances generalization across deepfake modalities.

The main contribution is the use of the S-transform on feature space to improve generalization beyond the training set to other deepfake modalities. The proposed method shows strong performance increases across several standard deepfake benchmarks.

**Audience:**

Yes

**Claims And Evidence:**

Yes

**Requested Changes:**

Please edit the paper to address the critiques mentioned in the above "weaknesses" section.

Number 3 above does not need to be applied to all benchmarks/experiments you have documented. I think it is sufficient to either 1) demonstrate it on a single relevant experiment 2) provide a citation that builds such a function and reports results on at least some of the datasets you're using.

**Strengths And Weaknesses:**

I really like that the idea is simple and intuitive. It would be easy to implement, and if its effectiveness is correctly represented by the benchmarks provided, it could prove useful in real world settings. I also appreciate the experiments section, which I think is diverse enough to establish the method’s potential usefulness outside the lab.

My main critiques are 1) formatting and cleanliness of figures and 2) the description of the method 3) a learned non-linear function as baseline 4) it is not clear which S-transform implementation was used. Specifically:

1) Figures

- Figure 1
    - is a bit sloppy, arrows are not symmetric and intersect with text
- Figure 2
    - is too small, needs to be larger, its unreadable at standard zoom levels
    - it is not clear how the numbers on the pie charts relate to the numbers provided in the description of the figure
- Figure 3
    - section 3 indicates the proposed algorithm uses “spatial-frequency decomposition” but the figure just says “feature transformation”
   the language should be aligned here for clarity
- Figure 4
    - would benefit from labelling the axis with the terminology from the methods section, i.e. y-axis is frequency (z) and the x-axis is k
    (“time”) i.e. feature vector from DenseNet
Figure 5
    - is off center and awkward, should be rotated so that its layed out horizontally, i.e. across the page rather than vertically down the page

There are more problems with many of the other figures, mostly formatting and text size being too small. Please provide better formatting and more legible font sizes to all tables and figures

2) Method

The end-to-end process is not clear enough, in my opinion. My guess that you’re taking keyframes from videos, ingesting via DenseNet, running the S-transform on features, and then combining the results somehow on a per video basis to form a decision about the video. There are a number of clarifications I think are required here:
    - It should be clearer that S-transform does not create a spatial decomposition per se, it simply provides a time-frequency decomposition, and the spatial component is implicit in the feature vector axis itself. I know this is obvious to the authors, but to people reading the paper who are not firmly embedded in the deepfake literature, it is going to be less clear
    - It is not formally stated what the processing steps are from the videos in the dataset to the network input, please clarify and formalize this, including any resizing / cropping / augmentation that may take place. Also, I believe that DenseNet ingests 224x224 images, please state this clearly if correct
    - it is not clear how the results are summarized – if key frames are taken, is each keyframe considered separately? or are these summarized after processing to form a decision on a per-video basis?

3) a learned non-linear function as baseline

Additionally, I think it would strengthen the paper if you compared against a baseline where a non-linear discrimination function is learned over feature space, in place of the proposed method. It is plausible that the S-transform / linear separator acts as a regularizer that a neural network would not be able to optimize for, but it would strengthen your argument if you proved that this was the case. I don’t see any mention of deepfake methods in the relevant work section that attempt to learn a function over feature space, if such methods exist adding them would be good as well.

4) It is not specified which implementation of the S-transform is used.

Brown et al. developed a fast, open source S-transform in ~2010, as per below references.

Brown, RA; Frayne, R (2008). "A fast discrete S-transform for biomedical signal processing". 2008 30th Annual International Conference of the IEEE Engineering in Medicine and Biology Society. Vol. 2008. pp. 2586–9.

Brown, Robert A.; Lauzon, M. Louis; Frayne, Richard (January 2010). "A General Description of Linear Time-Frequency Transforms and Formulation of a Fast, Invertible Transform That Samples the Continuous S-Transform Spectrum Nonredundantly". IEEE Transactions on Signal Processing. 58 (1): 281–290.

---

> ### Author Response · Authors · 2025-01-01
> **Response**
>
> **Editorial Changes:** We want to thank the reviewer for providing constructive feedback and helping to enhance the quality of the manuscript. We have carefully updated the figures and Tables to ensure the enhanced readability of the values and text.
>
> **Method:** We have updated the proposed algorithm to highlight that the s-transform does not perform spatial-frequency decomposition, it is a time-frequency decomposition technique, where in our content time is the feature representation obtained from the CNN.
>
> Similarly, the dataset section has been updated to mention the implementation of feature extraction and deepfake classification. For your quick reference text has been pasted here from the section:
>
> “To compute the image representation, faces are first cropped from the input images/videos and resized to a fixed size of 224x 224x3 and are normalized based on the eye coordinates. Apart from this standard processing of images, no other pre-processing has been applied to the face images. Since the number of frames in the existing deepfake videos is very high and they do not contain discriminating information at every frame; therefore, we have randomly selected 10 frames per video for detection. To classify, a video as deepfake or real, the average score of all the frames of the video is used as a score of that video. Such reduction of the frame is inspired by the current literature (Agarwal & Ratha (2024)) and also helps in reducing the computational cost while maintaining a significant deepfake detection performance.” The number of testing videos are same and defined in the dataset protocol, henceforth, the proposed results are easily comparable with the existing algorithm.
>
> **Non-Linear Classifier:** We want to highlight that there is no study exists that learned the non-linear discrimination function on top of the feature space. However, in some sense, we can say that the existing algorithm learned the non-linear discrimination function since the majority of them train the deep learning models end-to-end using non-linear activation functions.
>
> Based on the suggestion of the reviewer, we have performed the baseline study where we have trained two non-linear functions: (i) SVM with polynomial with degree 3 and RBF functions and (ii) 3-layer neural networks with a non-linear activation function. It is observed that the performance of these non-linear classifications under a cross-dataset (FF++ vs Celeb-DF) setting is at least 6% lower than the proposed linear approach. Moreover, the computational cost of training a non-linear is significantly higher than the proposed linear algorithm.
>
> **Implementation:** The implementation of S-transformation is based on the approach derived in the original paper titled Localization of the Complex Spectrum: The S Transform," introduced by R.G. Stockwell, L. Mansinha, and R.P. Lowe in 1996.  The implementation used the FFT with a Gaussian window. **We will release the source code used to perform the S-transform for reproducibility.**

---

> > ### Comment · Reviewer_U5rS · 2025-01-30
> > **Thank you**
> >
> > Thank you for your response, I've updated my review.
> >
> > Best regards,

---

### Review · Reviewer_Hu8M · 2025-01-01

**Summary Of Contributions:**

The paper focuses on the problem of deepfake multimedia detection and presents a novel, parameter-free and state-of-the-art algorithm. The proposed algorithm is evaluated on deepfake detection with different databases, different settings and different attacks and showing that the proposed algorithm is showing the best performance in all evaluated settings.

**Audience:**

Yes

**Broader Impact Concerns:**

The authors clearly discuss the broader impact of the proposed work in Sections 3.1 and 6.

**Claims And Evidence:**

Yes

**Requested Changes:**

**1. [important]** It can be beneficial to discover the cost-effectiveness of the proposed architecture to the existing methods.

**2. [important]** An ablation study can be beneficial to observe the effects of each component of the proposed architecture. In addition, the selection of CNN backbone and classifier can also be discovered.

**3.** For readability, some of the references needed to be fixed. Some references are given twice. For instance, in the second paragraph of the first page, a reference is given as "In one of the early works, Agarwal et al. Agarwal et al. (2017a)". In addition, the writing can be improved.

**Strengths And Weaknesses:**

## Strengths

1. The authors propose feature transformation based algorithm instead of the existing image transformation based algorithms to detect manipulated multimedia.
2. As the authors claim, it is the first work utilizing S-transform on the deepfake detection problem and the idea of utilizing S-transform is interesting.
3. The proposed architecture is outperforming the existing methods with different quality images on different attack settings.

## Weaknesses

1. The writing needs improvement. The paper is not easy to follow.
2. The proposed algorithm is claimed to be cost-effective but is not compared to the existing methods.

---

> ### Author Response · Authors · 2025-01-02
> **Response**
>
> **Cost:** Based on the suggestion, we have updated section 5.8 to discuss the cost-effectiveness of the proposed approach. Since the proposed approach does not have huge parameters to learn it is cost-effective as compared to the existing works. In other words, in the proposed approach, we have to optimize 2-3 parameters (of SVM) as compared to the millions of trainable parameters used in the existing algorithms utilizing deep learning architectures.
>
> **Ablation Studies:** Existing studies have demonstrated the effectiveness of various pre-trained models [A,B,C] and observed that DenseNet outperforms the other networks including VGG, Xception, and InceptionNet.
>
> However, based on the suggestion, we have performed two forms of ablation studies: (i) fine-tuning of pre-trained models for deepfake detection (can be seen as an evaluation of the first component of the proposed architecture) and second inclusion of second best architecture in our proposed architecture.
>
> The results reported in the updated Table 6 back the claim observed in the literature that the VGG performs poorly as compared to other pre-trained models. Further, it is observed that while the proposed architecture with any other CNN also yields state-of-the-art results, DenseNet yields the best performance. The slight reduction in performance with a change in the image representation architecture showcases that the proposed algorithm is agnostic to the architecture of deep networks used for image representation. It is to note here that while VGG alone performs poorly, it yields only 4-5% lower performance than the proposed algorithm with DenseNet when it is used in the proposed algorithm, **which shows the effectiveness of s-transformation of a feature space**. *In other words, the proposed algorithm with DenseNet image representation yields 4-5% better performance than the proposed algorithm VGG image representation.*
>
> [A] Masood M, Nawaz M, Javed A, Nazir T, Mehmood A, Mahum R. Classification of Deepfake videos using pre-trained convolutional neural networks. In IEEE 2021 International Conference on Digital Futures and Transformative Technologies (ICoDT2) 2021 May 20 (pp. 1-6).
>
> [B] A. Agarwal, N. Ratha, Deepfake Catcher: Can a Simple Fusion be Effective and Outperform Complex DNNs?, CVPR Workshop and Challenge on DeepFake Analysis and Detection (CVPRW), 2024, pp. 3791-3801
>
> [C] Zi B, Chang M, Chen J, Ma X, Jiang YG. Wilddeepfake: A challenging real-world dataset for deepfake detection. InProceedings of the 28th ACM International Conference on multimedia 2020 Oct 12 (pp. 2382-2390).
>
> **Citation Error:** Based on the suggestion of reviewers Hu8M and FXAY, we have updated references including the removal of the second occurrence.
>
> **Editorial Changes for Readability:** Apart from these suggestions, we have updated the manuscript based on the detailed comments of the other reviewers and thoroughly proofread it to improve the readability of the paper.

---

### Decision · Action_Editor_zZzT · 2025-02-21

**Recommendation:** Reject

**Comment:**

The manuscript lacks a rigorous analysis of the proposed method’s effectiveness beyond reporting accuracy improvements. There is limited justification for the choice of the S-Transform. Without comprehensive validation, its necessity and appropriateness remain unclear. Key claims are not fully supported, and the manuscript does not explain why the method improves performance. Some descriptions are difficult to follow, and figures (e.g., Figures 4, 7, and 9) lack clarity. Applying the S-Transform to CNN-extracted features, which are already highly abstract, raises concerns about its validity. If effective, it would be more natural to apply it directly to pixel space. A deeper analysis and additional experimental validation are needed.

**Audience:**

While deepfake detection is a relevant topic, the manuscript lacks a rigorous and well-supported analysis of the proposed method’s effectiveness beyond simply reporting accuracy improvements. The algorithm's design appears ad hoc, with little justification for the choice of the S-Transform. Without a comprehensive validation, it remains unclear whether this transformation is both necessary and appropriate for the task, making it difficult for readers to be convinced of its effectiveness.

**Claims And Evidence:**

The claims made in the submission are not fully supported by clear and convincing evidence. The manuscript lacks a thorough explanation of why the proposed method improves performance. Additionally, key descriptions remain difficult to understand, and some figures (e.g., Figures 4, 7, and 9) are unclear.

The authors state that they have applied a transformation to an image representation using the Stockwell Transform. However, it is important to consider that CNN-extracted image representations are already highly abstract. Applying further transformations to decompose spatial and frequency information in this feature space raises concerns about its meaningfulness and validity. The manuscript does not provide a convincing justification for why this approach is appropriate or beneficial.

Furthermore, if the Stockwell Transform is indeed effective, why not apply it directly in the pixel space of the image rather than on the extracted image feature space? A more comprehensive and in-depth discussion is needed to address this issue. Additionally, corresponding experimental results should be included to substantiate the proposed approach.

**Resubmission Of Major Revision:**

The authors may consider submitting a major revision at a later time.